# The Beneficial Effect of the COVID-19 Vaccine Booster Dose among Healthcare Workers in an Infectious Diseases Center

**DOI:** 10.3390/vaccines10040552

**Published:** 2022-04-02

**Authors:** Agata Skrzat-Klapaczyńska, Carlo Bieńkowski, Justyna Kowalska, Marcin Paciorek, Joanna Puła, Dominika Krogulec, Jarosław Stengiel, Agnieszka Pawełczyk, Karol Perlejewski, Sylwia Osuch, Marek Radkowski, Andrzej Horban

**Affiliations:** 1Department of Adults’ Infectious Diseases, Hospital for Infectious Diseases, Medical University of Warsaw, 02-091 Warsaw, Poland; carlo.bienkowski@gmail.com (C.B.); jdkowalska@gmail.com (J.K.); mpaciorek@zakazny.pl (M.P.); jpula@zakazny.pl (J.P.); dkrogulec@zakazny.pl (D.K.); ahorban@zakazny.pl (A.H.); 2Department of Adults’ Infectious Diseases, Hospital for Infectious Diseases, 02-091 Warsaw, Poland; jstengiel@zakazny.pl; 3Department of Immunopathology of Infectious and Parasitic Diseases, Medical University in Warsaw, 02-091 Warsaw, Poland; apawelczyk@wum.edu.pl (A.P.); kperlejewski@wum.edu.pl (K.P.); sosuch@wum.edu.pl (S.O.); mradkowski@wum.edu.pl (M.R.)

**Keywords:** healthcare workers, COVID-19, antibody response, vaccination

## Abstract

Introduction: Healthcare workers in Poland received a booster dose of the BNT162b2 mRNA vaccine (Pfizer-BioNTech, Manufacturer: Pfizer, Inc., and BioNTech; Moguncja, Germany) at the beginning of October 2021. Here, we report on the preliminary results of an ongoing clinical study into the antibody response to SARS-CoV-2 of healthcare workers previously exposed to the virus, with or without evidence of past infection, in the Hospital for Infectious Diseases in Warsaw before and after the vaccine booster dose. Methods: Blood samples were collected on the day the vaccine booster dose was administered and again 14 days later. The levels of SARS-CoV-2 IgG antibodies (against the n-protein, indicative of disease) and S-RBD (indicative of a response to vaccination) were measured. Results: One hundred and ten health care workers from the Hospital for Infectious Diseases were included in the study. The percentage of subjects with a positive test for anti-n-protein IgG antibodies at both time points remained unchanged (16, 14%), while a statistically significant increase in the percentage of subjects producing high levels of S-RBD antibodies (i.e., >433 BAU/mL) was observed (from 23, 21% to 109, 99%; *p* = 0.00001). Conclusions: The results of the study indicate that the booster dose of the vaccine significantly increases the percentage of people with high levels of S-RBD antibodies, regardless of previous contact with the virus, which may indicate greater protection against both the disease and a severe course of COVID-19.

## 1. Introduction

The novel coronavirus (SARS-CoV-2) responsible for coronavirus disease (COVID-19) was first detected in late 2019 in China [1]. SARS-CoV-2 belongs to the Betacoronaviridae, also in which other viruses that caused outbreaks in the past are also included (SARS-CoV and MERS-CoV). Within these viruses, mutations may occur not only due to frequent recombination, but also due to interspecies transmission [2]. Therefore, during the pandemic, a few new SARS-CoV variants formed, including alpha, beta, gamma, delta, and omicron [3]. The infection usually mildly affects the respiratory tract; however, it may also have a severe course with acute respiratory distress syndrome and multiple organ failure [4]. The first cases in Poland were diagnosed at the beginning of March 2020, and by January 2022, over 4,220,000 cases and 100,000 deaths had been reported [5]. On 27 December 2020, Poland introduced a mass vaccination program using the BNT162b2 mRNA vaccine (Pfizer-BioNTech) [6]. Healthcare workers (HCWs) were prioritized for COVID-19 immunization. The vaccine was given in two doses, three weeks apart. At the beginning of October 2021, a booster dose was administrated to fully vaccinated HCWs. It has been proven that anti-nucleocapsid antibodies serve as a marker of previous SARS-CoV-2 infections [7]. Anti-nucleocapsid antibodies can be also used as an indicator of post natural SARS-CoV-2 infection [8]. Meanwhile, a strong immune response to the virus’s spike protein, particularly the receptor-binding domain (RBD) of the spike protein (which contains neutralizing epitopes), is considered to be a response provoked by SARS-CoV-2 vaccines [9].

Here, we report the preliminary results of an ongoing clinical study on the effectiveness of a booster dose of the Pfizer mRNA vaccine among healthcare workers in the Hospital for Infectious Diseases in Warsaw.

## 2. Methods

The study participants were adults who had previously been vaccinated (8–9 months) with two doses of the BNT162b2 mRNA vaccine (Pfizer-BioNTech). Blood samples were collected on the day the vaccine booster doses were administered, and again 14 days later (October–November 2021) (Figure 1). The levels of SARS-CoV-2 IgG antibodies (against the n-protein, indicative of disease) and S-RBD antibodies (indicative of a response to vaccination) were measured using MAGLUMI SARS-CoV-2 IgG and MAGLUMI SARS-CoV-2 S-RBD IgG assays. According to the manufacturer’s information, MAGLUMI^®^ SARS-CoV-2 S-RBD IgG kits are 99.6% specific and 100% sensitive. The kits have been approved for sale in the European Union and have received a CE certificate.

The group of COVID-19-recovered participants was distinguished based on positive PCR test results for SARS-CoV-2 from any time prior to the booster dose.

Data on concomitant diseases were collected on the basis of a survey conducted among the study participants.

In the statistical analyses, non-parametric tests were used as appropriate: the Chi2 test to compare categorical variables and the Wilcoxon test to compare dependent numeric variables.

A *p* value of <0.05 was considered significant.

The study was approved by the Bioethical Committee of the Medical University of Warsaw (Nr KB/2/2021). The study was funded from a research grant issued by the Medical Research Agency (Nr 2021/ABM/COVID19/WUM).

## 3. Results

One hundred and ten healthcare workers from the Hospital for Infectious Diseases in Warsaw were included in the study. Most participants were female (87, 79.1%) and were working in direct contact with patients (83, 74.5%). In terms of professions, there were 31 doctors (28.2%), 21 nurses (19.1%), and 31 from other professions (28.2%). Their median height was 1.65 m (IQR: 1.6–1.73 m), weight 70 kg (IQR: 61–84 kg), and BMI 25.15 kg/m^2^ (IQR: 22.86–29.30 kg/m^2^). Twenty-three participants (20.9%) had at least one concomitant disease. Concomitant diseases were more common in the group of COVID-19-recovered participants (34.6% vs. 16.7%, *p* = 0.0492). As for the antibody titers, our analysis revealed that the median level of IgG antibodies on the day that the booster dose was administered was 0.0575 AU/mL (IQR: 0.0170–0.21975 AU/mL), and the S-RBD antibodies’ median titer was 159.2358 BAU/mL (IQR: 70.8776–394.6254). Two weeks after receiving the vaccine booster dose, these antibody titers had changed: the median IgG was 0.0855 AU/mL (IQR: 0.0320–0.2600), and the S-RBD reached the maximum value possible to detect in the laboratory (median 433 BAU/mL, IQR: 433–433 BAU/mL) (Table 1).

Twenty-six healthcare workers (23.6%) had a SARS-CoV-2 infection confirmed before receiving the booster dose of BNT162b2 (Comirnaty) vaccine and were considered COVID-19-recovered participants before the booster vaccine dose. Their characteristics are presented in Table 1.

Our analysis revealed that the presence of at least one concomitant disease was more likely to occur in the COVID-19-recovered group in our hospital. Within this group, two people had three concomitant diseases concurrently (hypertension, type II diabetes mellitus, and a history of myocardial infarction; the second person had asthma, hypertension, and was undergoing immunosuppressive treatment due to an interstitial lung disease), and seven people had one concomitant disease (hypertension (five people), hyperthyroidism, and one was a bone marrow transplant recipient).

As for the group of subjects who had not had COVID-19 in the past, two participants had more than one disease (hypertension and asthma, and the second person had hypertension and a history of myocardial infarction); the rest had hypertension (seven people), immunosuppressive treatment, asthma, obesity, hypothyroidism, or chronic hepatitis.

The percentage of healthcare workers with a positive test for anti-n-protein IgG antibodies at both time points did not differ (16, 14%), while a statistically significant increase in the percentage of people with very high levels of S-RBD antibodies (i.e., >433 BAU/mL) was observed. On the day the booster dose of the vaccine was administered, 23 (20.9%) of the subjects had S-RBD antibodies > 433 BAU/mL, while two weeks later, 109 (99%) of the subjects had the antibodies, *p* = 0.00001 (Table 2, Figure 2).

## 4. Discussion

Recent studies have shown the short-term efficacy of a two-dose regimen of BioNTech/Pfizer mRNA BNT162b2 vaccine against COVID-19. Analyses have confirmed this efficacy in both clinical trials and real-world settings, using a two-dose schedule with a target interval of three weeks between doses [6,10]. On the other hand, there is evidence for waning SARS-Co-V-2 immunity after a few months of receiving the second vaccine dose in Israel. After a successful vaccination campaign starting in December 2020, more than half the adult population received two doses of BNT162b2 vaccine in three months. In June 2021, an increase in the number of SARS-CoV-2 PCR tests was observed among both vaccinated and unvaccinated persons. Genetic analyses showed that the delta variant was responsible for most cases (98%) of the observed breakthrough infections during that time [11,12]. Reports of waning vaccine efficacy came from healthcare organizations all over the world [13,14,15]. In addition, there was a decrease in neutralizing antibody titers during the first six months after receiving the second dose of the vaccine [16].

Recent analyses have suggested that a booster dose of the BNT162b2 vaccine reduced the rates of both confirmed infection and severe COVID-19 illness [17]. The rate of confirmed infection was significantly lower than in the non-booster group. Moreover, in a study by Arbel et al., participants who received a booster dose of BNT162b2 at least five months after the second dose had 90% lower mortality rates due to COVID-19 than participants who did not receive a booster dose of the vaccine [18]. These two above studies reported no side effects during the period of the research. The protection gained by a booster dose is relevant for public health, especially in the context of waning vaccine efficacy. The booster dose and the expected reduction in the incidence and severe course of COVID-19 highlight the important role of maintaining immunity in the general population, which is critical for public health, especially in terms of saving lives. Study reports show that the vaccine’s efficacy against primary symptomatic COVID-19 was directly related to an increase in the anti-SARS-CoV-2 receptor-binding domain (RBD) IgG; these antibodies also appear to correlate better with virus neutralization [19]. In their study, Carta et al. strongly underlined that only serology assays specific for antibodies that target regions within the spike protein (e.g., the RBD) can be used to evaluate immune response to the BNT162b2 vaccine [20]. Moreover, other studies also supported these findings, stating that antibodies only against the spike protein will be elicited after the BNT162b2 vaccine [21].

In our study, a significant increase in the percentage of people with very high levels of anti-S-RBD antibodies (>433 BAU/mL) was observed after a booster dose of the BNT162b2 mRNA vaccine (Pfizer-BioNTech).

Narayan et al. analyzed data from 20 different hospitals and collected a study group of 14,837 HCWs. They underlined that vaccination against COVID-19 is effective. However, their analysis included HCWs vaccinated with only two vaccines doses [22]. Poukka et al. also analyzed two-vaccine dose efficacy in HCWs and also concluded that a vaccine booster dose may be beneficial for COVID-19 prevention in HCWs [23]. However, the efficacy of a booster dose in HCWs should be analyzed, due to the fact that the risk of exposure for COVID-19 in the workplace is higher for HCWs than for general population [24].

In our analyses, there was a statistically significant increase in the titer of S-RBD antibodies when comparing the values before and after the vaccine booster dose (*p* = 0.00014) among COVID-19-recovered participants. RBD-specific antibodies with strong antiviral activity were found in studies with COVID-19-recovered participants in the pre-vaccine period, suggesting that a vaccine designed to raise such antibodies could be very effective [25]. This was the reason high expectations were placed on vaccines.

However, the data show that the COVID-19 vaccine can elicit specific antibody titers and neutralizing antibody concentrations above those observed in COVID-19 human convalescent serum in the first 100 days after vaccination [26]. Further analyses are needed in this area.

Our study was conducted among healthcare workers, who are generally healthy and relatively young. This could be the limitation of our study, as this group of people was not compared with the general population.

Despite these limitations, our findings support the recommendation of providing the COVID-19 vaccine booster dose to the broad population. HCWs were prioritized for COVID-19 immunization in our country, and it was a natural choice for obtaining initial results quickly. To our knowledge, this is the first study to evaluate beneficial effects of the booster dose based on the immune response in the Polish population.

## 5. Conclusions

The preliminary results of our study indicate that the vaccine booster dose significantly increases the percentage of people with high levels of anti-SARS-CoV-2 S-RBD IgG antibodies. It may indicate better protection against both the disease and a severe course of COVID-19, regardless of previous contact with the virus. Future research is needed to evaluate the long-term efficacy of the booster dose against current and emerging SARS-CoV-2 variants.

## Figures and Tables

**Figure 1 vaccines-10-00552-f001:**
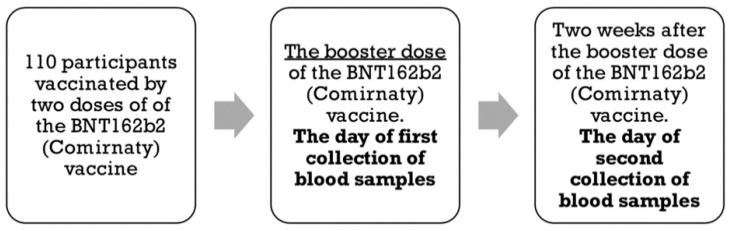
The scheme showing the steps of the study.

**Figure 2 vaccines-10-00552-f002:**
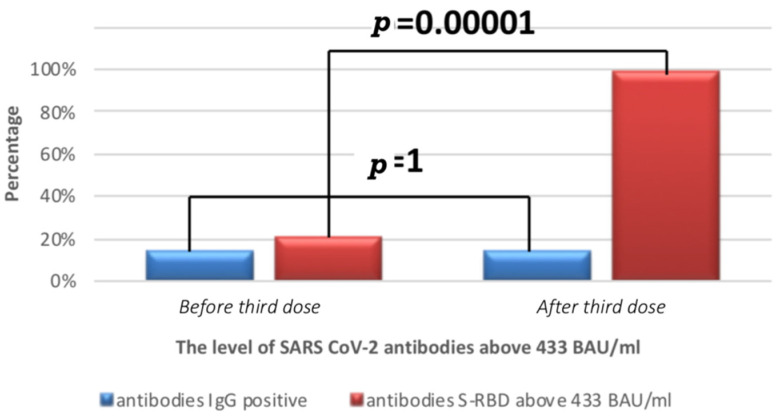
Percentage of people with anti-n IgG and S-RBD antibody levels on the day of administration of the third dose of anti-COVID-19 vaccine and 14 days after administration of the third dose of anti-COVID-19 vaccine.

**Table 1 vaccines-10-00552-t001:** Baseline characteristics stratified by COVID-19 infection status before vaccination with a booster dose of the BNT162b2 (Comirnaty) vaccine.

Characteristic	All (N = 110)	COVID-19-Recovered Subjects (N = 26)	Subjects without COVID-19 in the Past (N = 84)	*p*-Value
Age in years, median [IQR *]	51.0 [39.5–58.5]	51.5 [46.0–57.5]	49.0 [38.0–59.0]	0.7188 ^a^
Female sex, *n* (%)	87 (79.1)	21 (80.8)	66 (78.6)	0.8097 ^b^
BMI ** in kg/m^2^, median [IQR]	25.15 [22.86–29.30]	25.10 [23.60–31.24]	25.39 [22.70–28.75]	0.9920 ^a^
One or more concomitant disease, *n* (%)	23 (20.9)	9 (34.6)	14 (16.7)	0.0492 ^b^
Working directly with patients, *n* (%)	83 (75.5)	22 (84.6)	61 (72.6)	0.2414 ^b^
IgG *** in AU/mL on the day the booster vaccine dose was given, median [IQR]	0.0575 [0.0170–0.21975]	0.0580 [0.0250–0.0733]	0.0525 [0.0140–0.1638]	0.4715 ^a^
S-RBD **** in BAU/mL on the day the booster vaccine dose was given, median [IQR]	159.2 [70.9–394.6]	175.8 [111.7–426.7]	149.6 [58.3–348.0]	0.8887 ^a^
S-RBD > 433 BAU/mL 2 on the day the booster vaccine was given, *n* (%)	23 (20.9)	7 (15.4)	16 (19.1)	0.3881 ^b^
IgG 2 in AU/mL 2 weeks after the booster vaccine dose, median [IQR]	0.0855 [0.0320–0.2600]	0.1490 [0.0400–0.9168]	0.0840 [0.0288–0.2423]	0.5419 ^a^
S-RBD > 433 BAU/mL 2 weeks after the booster vaccine dose, *n* (%)	107 (97.27)	26 (100)	83 (98.81)	0.3758 ^b^

* IQR—interquartile range; ** BMI—body mass index; *** IgG—antibodies against the SARS-CoV-2 in IgG class; **** S-RBD—antibodies anti-spike protein receptor-binding domain. ^a^ Chi2 test; ^b^ Wilcoxon test.

**Table 2 vaccines-10-00552-t002:** SARS-CoV-2 serological tests among HCWs from the Hospital for Infectious Diseases (Warsaw, Poland) before and after the booster dose of the BNT162b2 (Comirnaty) vaccine.

	Before the Booster Dose	2 Weeks after the Booster Dose	*p*-Value
Qualitative			
Positive anti-SARS-CoV-2 IgG, *n* (%)	16 (14%)	16 (14%)	1.0000 ^a^
Positive anti-SARS-CoV-2 S-RBD > 433 BAU/mL, *n*(%)	23 (20.9%)	109 (99.1%)	<0.0001 ^a^
Quantitative			
Anti-SARS-CoV-2 IgG, Median (IQR)	0.0575 [0.0170–0.21975]	0.0855 [0.0320–0.2600]	0.0424 ^b^

Anti-SARS-CoV-2 antibodies MAGNUMI platform IgM index ≥ 1.10 positive; IgG ≥ 1.00 AU/mL is positive; S-RBD ≥ 4.33 BAU/mL is positive; ^a^ Chi2 test; ^b^ Wilcoxon test.

## Data Availability

The data presented in this study are available on request from the corresponding author.

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
