# Peer review of "The Beneficial Effect of the COVID-19 Vaccine Booster Dose among Healthcare Workers in an Infectious Diseases Center"

_vaccines, 2022, doi:10.3390/vaccines10040552_

Round 1

Reviewer 1 Report

In the paper, the authors show that the booster dose of the vaccine significantly increases the percentage of people with high levels of S-RBD antibodies, regardless of previous contact with the virus, which may indicate greater protection against both the disease and a severe course of COVID-19.

Suggestions for improvement:

-The paper should be accompanied by graphs showing the main results, besides the Tables.

-When quantifying IgG response to the booster dose of the vaccine, a paired statistical analysis should be performed

-Please double-check the citation and order of Tables 1 and 2

Reviewer 2 Report

The authors evaluate in this sort article the effect of COVID-19 vaccine booster dose on a population of Healthcare workers. They show by antibody titration, before and after booster injection, a significant increase of S-RBD antibodies. Though the population is mostly women aged around 50, this study show that after a waning of immunity after two vaccine injections in this group, booster injection re-establishes a high level of level of COVID-19 immunity.

There are some minor points:

- in abstract line 23, the number "109" should not appear in "from 23.21% to 99% p=0.00001)

- paragraph mentioning COVID-18 recovered participants (lines 99-102) should be inserted before, perhaps before line 82.

- there is an increase of anti SARS-CoV-2 IgG against n-protein (p-value 0.0424) though the number of positive case doesn't change during the study. Though mentioned in the text (line 79) that there is a significant change, do the authors have a hypothesis for this increase?

Author Response

Reviewer 2

The authors evaluate in this sort article the effect of COVID-19 vaccine booster dose on a population of Healthcare workers. They show by antibody titration, before and after booster injection, a significant increase of S-RBD antibodies. Though the population is mostly women aged around 50, this study show that after a waning of immunity after two vaccine injections in this group, booster injection re-establishes a high level of level of COVID-19 immunity.

There are some minor points:

- in abstract line 23, the number "109" should not appear in "from 23.21% to 99% p=0.00001)

We would like to explain, that there is a sentence: “The percentage of subjects with a positive test for anti-n-protein IgG antibodies at both time points remained unchanged (16, 14%), while a statistically significant increase in the percentage of subjects producing high levels of S-RBD antibodies (i.e. >433 BAU/ml) was observed (from 23, 21% to 109, 99%; p=0.00001). It means that 23 and 109 are numbers of subjects and 21 and 99 are percents from 110 subjects

- paragraph mentioning COVID-18 recovered participants (lines 99-102) should be inserted before, perhaps before line 82.

This has been corrected

- there is an increase of anti SARS-CoV-2 IgG against n-protein (p-value 0.0424) though the number of positive case doesn't change during the study. Though mentioned in the text (line 79) that there is a significant change, do the authors have a hypothesis for this increase?

Thank you for this question. The IgG antibodies increase after the BNT162b2 vaccine booster dose issignificant. Nonetheless, the minimum titer that may be interpreted as positive result is 1 AU/mL.Therefore, the qunatitative increase is significant, but the qulitative result is not. The increase is vaccination-induced, although clinically may be less significant.

Reviewer 3 Report

In this article, the authors deal with the beneficial effect of the COVID-19 vaccine booster dose among healthcare workers in an infectious diseases centre.

The manuscript is interesting, but there are still some points to be addressed by the authors in order to improve their manuscript.

A scheme with all the steps of this study should be included

The introduction section is too short and has to be included more details about SARS-CoV-2, its mutations and few clinical data.

Line 37. What previous SARS-CoV-2 infections do the authors refer to? Add more details about these studies. A correlation between anti-nucleocapsid antibodies and previous SARS-CoV-2 infections mentioned by the authors should also be made.

Discussion section:

-The novelty of the study and its clinical importance are not highlighted.

-Mention the limitations of this study.

-What is the importance of this study for public health? Explain why the study was done only on subjects working in the medical sector and highlight the differences compared to the rest of the population. SARS-CoV-2 infection is a global public health problem.

- Lines 112-113: Define the terms and immunologically argue the statement: "two-dose regimen of BioN-Line Tech / Pfizer mRNA BNT162b2 vaccine against COVID-19. References 6 and 7 do not support this statement. Revise it.

- Lines 114-115: Ref 8 concluded that immunity against the delta variant of SARS-CoV-2 waned a few months after receipt of the second dose of vaccine in Israel. Should readers understand that all of these subjects were infected with the Delta variant? Add additional data if about the genomic sequencing of the positive subjects analyzed.

Lines 119-128: mention if the mentioned studies also have side effects or not.

Lines 133-136: Add more studies from the literature for comparison with this study.

Line 137: This statement is not clear.  Mention from which point of view the efficacy of the booster dose in HCWs should be analyzed.

The conclusion section is missing. Do the authors not have relevant conclusions for this research of theirs?

Author Response

Reviewer 3
In this article, the authors deal with the beneficial effect of the COVID-19 vaccine booster dose among healthcare workers in an infectious diseases centre.

The manuscript is interesting, but there are still some points to be addressed by the authors in order to improve their manuscript.

A scheme with all the steps of this study should be included

A scheme has been added as a Figure 1 named “The scheme showing the steps of the study” in the Methods section

The introduction section is too short and has to be included more details about SARS-CoV-2, its mutations and few clinical data.

This has been added in the introduction section, lines 31-38. We’ve added also references 2,3 and 4.

Line 37. What previous SARS-CoV-2 infections do the authors refer to? Add more details about these studies. A correlation between anti-nucleocapsid antibodies and previous SARS-CoV-2 infections mentioned by the authors should also be made.

We mean that nucleocapsid antibodies are the standard assay for the detection of previous infection which means that they can be also used as an indicator of post natural SARS CoV-2 infection.

This has been added in the Introduction section, lines 45-46 with the new reference nr 8.

Discussion section:

-The novelty of the study and its clinical importance are not highlighted.

It has been added in the lines: 215-223, also lines 174-179

-Mention the limitations of this study.

The is a paragraph mentioned in the Discussion section: “Our study was conducted among healthcare workers, who are generally healthy and relatively young. This could be the limitation of our study, as this group of people hasn’t been compared with the general population.”

-What is the importance of this study for public health? Explain why the study was done only on subjects working in the medical sector and highlight the differences compared to the rest of the population. SARS-CoV-2 infection is a global public health problem.

It has been added in the lines: 175-179

- Lines 112-113: Define the terms and immunologically argue the statement: "two-dose regimen of BioN-Line Tech / Pfizer mRNA BNT162b2 vaccine against COVID-19. References 6 and 7 do not support this statement. Revise it.

It has been defined. The one refence has been changed.

- Lines 114-115: Ref 8 concluded that immunity against the delta variant of SARS-CoV-2 waned a few months after receipt of the second dose of vaccine in Israel. Should readers understand that all of these subjects were infected with the Delta variant? Add additional data if about the genomic sequencing of the positive subjects analyzed.

It has been added in the discussion section, lines 161-166. The reference was also added.

Lines 119-128: mention if the mentioned studies also have side effects or not.

It has been added in the Discussion section lines: 174-175

Lines 133-136: Add more studies from the literature for comparison with this study.

It has been added, with the new reference in lines:185-186

Line 137: This statement is not clear.  Mention from which point of view the efficacy of the booster dose in HCWs should be analyzed.

It has been explained in the lines: 196-198. The new reference was added to support the statement.

The conclusion section is missing. Do the authors not have relevant conclusions for this research of theirs?

The Conclusion section has been added:

Conclusions

The preliminary results of our study indicate that the vaccine booster dose significantly increases the percentage of people with high levels of anti-SARS-CoV-2 S-RBD IgG antibodies. It may indicate better protection against both the disease and a severe course of COVID-19, regardless of previous contact with the virus. The future research is needed to evaluate long-term efficacy of the booster dose against current and emerging SARS CoV-2 variants.

Round 2

Reviewer 3 Report

No answer given.